# Validation of Pretreatment Prognostic Factors and Prognostic Staging Systems for Small Cell Lung Cancer in a Real-World Data Set

**DOI:** 10.3390/cancers14112625

**Published:** 2022-05-25

**Authors:** Raphael Hagmann, Alfred Zippelius, Sacha I. Rothschild

**Affiliations:** 1Division of Medical Oncology, University Hospital Basel, 4031 Basel, Switzerland; raphael.hagmann@gmx.ch (R.H.); alfred.zippelius@usb.ch (A.Z.); 2Lung Cancer Center Basel, Comprehensive Cancer Center, University Hospital Basel, 4031 Basel, Switzerland

**Keywords:** small cell lung cancer, prognostic factors, prognostic staging system, survival

## Abstract

**Simple Summary:**

We present an analysis of a real-world cohort of patients with small cell lung cancer (SCLC) and examine the value of prognostic factors and scores that have been published in recent decades. In our analysis, only a few clinical (age, tumor stage) and a single laboratory parameter (alkaline phosphatase) are associated with the prognosis of patients with SCLC. We could not confirm the prognostic role of most of the published complex prognostic scores.

**Abstract:**

Treatment decisions in patients with small cell lung cancer (SCLC) are made based on the extent of the disease. However, the outcome varies among patients at the same stage. A simple tool to predict outcomes in SCLC patients would be helpful for clinical decision-making. In recent years, several prognostic scores have been proposed. In this study, we evaluated the different prognostic factors in an unselected real-world cohort of patients. We retrospectively collected clinical, radiological and laboratory data from 92 patients diagnosed with SCLC. Univariate and multivariate cox regression analyses of survival were performed to assess the prognostic value of relevant clinical and laboratory factors for SCLC. Furthermore, we examined the association between eight published prognostic scores for SCLC and overall survival (OS). In the overall cohort, the median OS was 10.3 months (20.9 months and 9.2 months for limited disease (LD) SCLC and extensive disease (ED) SCLC, respectively). In univariate analysis, initial staging, number of metastatic sites and presence of liver, bone and adrenal gland metastases were significantly associated with worse OS. Of the established laboratory markers, albumin, alkaline phosphatase and hyponatremia but not lactate dehydrogenase (LDH) significantly predicted OS. All published prognostic scores, with the exception of the Glasgow Prognostic Score, did not significantly predict OS. In multivariate analysis, age, staging and alkaline phosphatase serum levels showed significant association with OS. We could not confirm the prognostic significance of most of the published complex prognostic scores. We therefore recommend using simple clinical and laboratory factors instead of complex scores to estimate the prognosis of SCLC patients in clinical practice.

## 1. Introduction

Lung cancer remains the leading cause of cancer death worldwide. Approximately 15% to 20% of lung cancers are of small cell histology. Small cell lung cancer (SCLC) has a very aggressive course, with a median overall survival (OS) of approximately 12–20 months, with only 6% to 12% of patients surviving 5 years after diagnosis, even in the early stages of the disease [1,2]. Most patients with SCLC have a history of smoking. The traditional staging system developed by the Veteran’s Administration Lung Cancer Study Group divides patients into two stages according to the extent of the disease [3]. Limited disease (LD) is confined to one hemithorax with regional lymph node metastasis and can be treated with a single radiation port. Approximately 60% to 70% of patients have extensive disease (ED) at initial diagnosis with a median survival of 7 to 11 months, and only 1% are alive at 5 years [4,5,6]. Patients with LD-SCLC are treated with concurrent chemo-radiotherapy (CRT), whereas patients with ED-SCLC are traditionally treated with platinum-based chemotherapy only [7]. First-line chemotherapy with cisplatin or carboplatin in combination with etoposide has been the standard of care [8] until recently. Additional therapy with a PD-L1 inhibitor (atezolizumab or durvalumab) has been shown to improve OS compared with chemotherapy alone [9,10] and is therefore a new standard of care. Overall, the prognosis of patients with ED-SCLC remains poor. Despite the curative potential in patients with LD-SCLC with combined CRT and high response rates with chemo(-immuno)therapy in patients with ED-SCLC, most patients relapse after a short time. Relapses are usually associated with a poor prognosis [11]. In a study by our group, we demonstrated for a real-world patient cohort that second-line chemotherapy in this setting leads to a response again in a substantial proportion of patients, but the overall prognosis is very unfavorable [12].

The identification of prognostic markers is important for the advancement of therapy in SCLC, as well as for everyday treatment decisions and discussions with the patient. Although newer biomarkers have recently been investigated in SCLC and their independent prognostic role has been demonstrated [13,14,15], they are not suitable for daily clinical care of patients with SCLC due to the complexity of determination, time and cost. Therefore, the establishment and validation of simple and cost-effective biomarkers that can be universally used in everyday clinical practice and integrated into the work-up and treatment algorithm are of interest. Several older studies have identified patient- and tumor-associated characteristics and laboratory parameters as prognostic factors in a patient with SCLC [16,17]. In recent years, there has been a great interest in inflammatory biomarkers and the establishment of prognostic scores [18,19,20]. The extent to which such parameters can predict prognosis in a real-world population of patients with SCLC and thus influence treatment decisions has not been extensively studied so far. Therefore, the aim of this study is to validate clinical and laboratory parameters as well as established prognostic scores in an unselected real-world population of patients with SCLC.

## 2. Materials and Methods

### 2.1. Patients

We identified 92 patients with SCLC diagnosed at the University Hospital of Basel between January 2000 and December 2010 by searching the patient database of the Department of Medical Oncology at the University Hospital Basel. Clinical, laboratory and therapeutic parameters were collected retrospectively from the hospitals’ electronic database and the patients’ medical records. The trial was approved by the cantonal ethics committee (EKBB, Ethical Committee of both Cantons Basel).

### 2.2. Staging, Response and Outcome Evaluation

Tumor stage at initial diagnosis was categorized according to the American Joint Committee on Cancer (AJCC) TNM staging system, 7th edition [21]. Additionally, we categorized patients into “limited disease” (LD) and “extensive disease” (ED) as defined by the IASCL consensus conference [22]. Overall response rate (ORR) was defined according to Response Evaluation Criteria in Solid Tumors (RECIST) criteria version 1.0 [23]. OS was defined as the time from diagnosis to death from any cause.

### 2.3. Parameters Included in the Univariate and Multivariate Analyses

We divided the studied parameters into clinical factors, tumor-specific factors, laboratory parameters, and therapy-associated factors. We included the following clinical factors in the analysis: age, gender, smoking status, European Cooperative Oncology Group (ECOG) performance status (PS), weight and weight loss. As tumor-specific factors, we recorded stage (LD vs. ED), number of metastatic sites, and specific metastatic sites (liver, brain, bone, adrenal glands, pleura). Analysis of laboratory parameters included sodium, albumin, alkaline phosphatase (AP), lactate dehydrogenase (LDH), bicarbonate, C-reactive protein (CRP), neutrophils and leukocytes at diagnosis, as well as the occurrence of hyponatremia in the first cycle of chemotherapy, calcium elevation in the first cycle of chemotherapy and the occurrence of hypercalcemia throughout the course of the disease. In addition, we included different treatment approaches as well as the response to therapy and the use of prophylactic cranial irradiation (PCI) as treatment-associated factors in the analysis.

### 2.4. Analysis of Established Prognostic Scores

We validated the following eight prognostic scores in our cohort: Advanced Lung Cancer Inflammation Index (ALI) [24], Glasgow Prognostic Score (GPS) as well as modified GPS (mGPS) [18,25], London Group Score [26], Manchester scoring system [17], neutrophil-lymphocyte rate (NLR) [20,27], new prognostic index (NPI) [16] and prognostic nutritional index (PNI) [28].

### 2.5. Statistics

Univariate survival analysis was performed using the Kaplan–Meier method, and statistical significance was assessed using the log rank test. Multivariate analysis was performed with the Cox regression model. All variables that reached statistical significance in the univariate analysis were included in the multivariate analysis. Patients who had missing information on a variable were excluded from analysis for this specific variable. A significance level of *p* < 0.05 was used for all tests. SPSS statistical software (IBM Corporation, New York, NY, USA) was used for all analyses.

## 3. Results

### 3.1. Patient Characteristics

We identified 92 patients with SCLC diagnosed between January 2000 and December 2010 at our institution. The median age at initial diagnosis was 63 years (range, 39.7–81.9 years). Of the 92 patients, 62 patients (67.4%) were male, and 22 patients (23.9%) were initially diagnosed with LD-SCLC. In the whole cohort, the median OS was 10.3 months. At the time of data cut-off, 83 patients were deceased, and four patients were still alive. Five patients were lost to follow-up and were therefore not included in the outcome analysis. Of those alive, three patients were initially diagnosed with LD-SCLC, and one patient was diagnosed with ED-SCLC. Of the patients with LD-SCLC, two patients underwent combined radio-chemotherapy (RCT), and one patient underwent chemotherapy and a surgical procedure and did not relapse. The patient with ED-SCLC was treated with four cycles of carboplatin and etoposide without additional radiotherapy, resulting in a radiographic partial response (PR), and now has more than 8 years of follow-up.

### 3.2. Univariate Analysis

A total of 32 parameters were included in the univariate analysis. These parameters were grouped in clinical and tumor-specific parameters (n = 13). The results of univariate analysis of clinical parameters are shown in Table 1.

In addition to stage (LD vs. ED, *p* < 0.001; Figure 1A), the number of metastatic sites (0–1 vs. ≥ 2, *p* < 0.001; Figure 1B) and the presence of metastases in the liver (*p* < 0.001; Figure 1C), bone (*p* < 0.001; Figure 1D) and adrenal glands (*p* = 0.028; Figure 1E) were statistically significantly associated with prognosis. In our cohort, 96.7% of patients were current or former smokers with a median nicotine consumption of 50 pack years (py). Using the median value as a cut-off, we see no prognostic impact of nicotine use. Gender (*p* = 0.061) and presence of brain metastases (*p* = 0.067) was of borderline significance in our analysis. Analysis of a total of 11 laboratory parameters is summarized in Table 2.

Hyponatremia (<131 mmol/L) at baseline (*p* = 0.041; Figure 2A) or during the first cycle of therapy (*p* = 0.012, Figure 2B) as well as hypalbuminemia (<35 g/L) (*p* = 0.044; Figure 2C) or elevated AP (≥129 U/L) (*p* < 0.001; Figure 2D) were significantly associated with a worse prognosis. Elevated LDH level (*p* = 0.068) was of borderline significance. Furthermore, we examined eight treatment-associated factors (Table 3).

Choice of first-line therapy (*p* < 0.001; Figure 3A) and response to first-line therapy (*p* < 0.001; Figure 3B) were statistically significantly associated with prognosis. The use of PCI (*p* = 0.073) was of borderline significance.

Of the eight prognostic staging systems selected, only GPS (*p* = 0.044; Figure 4) and the Manchester Score were significantly associated with outcome in univariate analysis (Table 4).

The London Group and the Manchester scoring system were calculated using historical and updated laboratory reference values. For the Manchester Score, prognostic significance was shown with both the historical (*p* = 0.008) and updated reference values (*p* = 0.012). For the London Group Score, as well as for the other prognostic scores, we did not find a statistically significant correlation with OS in univariate analysis. On the border of statistical significance were the London Group Score with updated thresholds and the prognostic nutritional index (PNI). Kaplan–Meier graphs of those factors that are significantly associated with overall survival in the univariate analyses are shown in Figure 1, Figure 2, Figure 3 and Figure 4.

### 3.3. Long-Term Survival

Baseline characteristics were also analyzed according to their relationship to long-term survival (Table 5, Table 6 and Table 7).

The cut-off for the definition of long-term survival was two years. Initial staging (*p* < 0.001), number of metastatic sites (*p* < 0.001), liver metastases (*p* = 0.007), brain metastases (*p* = 0.037), bone metastases (*p* = 0.014), hyponatremia (*p* = 0.037), albumin level (*p* = 0.046), AP (*p* = 0.049), type of first-line treatment (*p* < 0.001) and response to initial chemotherapy (*p* < 0.001) were significantly associated with long-term survival.

### 3.4. Multivariate Analysis

Since variables could covariate positively or negatively and therefore influence other variables in their prognostic importance, we performed a multivariate analysis with the significant variables from univariate analysis (Table 8).

We excluded the presence of liver, bone and adrenal gland metastases for multivariate analysis because they are already included in the variable number of metastatic sites. In addition, we excluded type of first-line therapy and response to initial therapy, as these are not baseline characteristics, and these factors are relevantly dependent on baseline characteristics. GPS was also excluded because it would be covariate with AP serum levels. In multivariate analysis, we found a statistically significant correlation between age (*p* = 0.018), initial tumor stage (*p* = 0.022) and AP serum level (*p* = 0.004).

## 4. Discussion

In this retrospective study, we investigated the significance of various clinical, laboratory and treatment-associated factors as well as established prognostic scores for OS of patients with SCLC. In this real-world analysis of unselected patients treated over an 11-year period at an academic center in Switzerland, we demonstrated that few clinical and laboratory parameters were associated with prognosis. In multivariate analysis, only age, initial tumor stage, and serum levels of AP were significantly associated with OS. Some other parameters, such as the number of metastatic sites; the presence of metastases in liver, bone and adrenal glands; and blood sodium and albumin levels, showed correlation with patient survival in univariate analysis. Thus, our study differs in several respects from previously published analyses on prognostic parameters. In numerous older studies, ECOG PS and gender were significantly associated with prognosis [29,30,31]. However, some conflicting data were found, and the influence of gender on prognosis seems to depend on the stage in particular. In a recently published real-world analysis, the female gender was significantly associated with prognosis only in patients with LD-SCLC [32].

Surprisingly, when we validated prognostic scores that have shown a correlation with prognosis in previous publications, we could not demonstrate an independent and statistically significant correlation between most of them with prognosis in our cohort. In univariate analysis, only the GPS and Manchester score were associated with survival. 

The Manchester Score is one of the oldest prognostic scores established in patients with SCLC [17]. The original study enrolled 407 patients between 1979 and 1985 and examined 61 parameters in a Cox multiple regression analysis. Patients were treated with chemotherapy (cyclophosphamide, etoposide, and methotrexate or ifosfamide and etoposide) in four different trials and received additional thoracic radiotherapy in case of complete remission after chemotherapy. Although the therapy of the patients in our study, which was conducted about 20 years later, was different, and all patients received platinum-based chemotherapy, we were able to confirm the prognostic significance of the Manchester score, the parameters of which are determined before therapy and the significance of which is apparently independent of the treatment of the patients.

The prognosis of tumor patients depends not only on tumor characteristics but also on patient-related factors that impact response to therapy. In recent years, it has become evident that tumor-associated inflammation is a relevant factor for tumor development and progression and thus the prognosis of tumor patients [33]. Furthermore, several studies have shown that inflammatory markers are relevant prognostic factors independent of tumor stage and clinical characteristics [34]. The use of inflammation scores incorporating various inflammatory markers reflects the complexity of the tumor-associated inflammatory response. Such scores have been established as valuable prognostic factors in several tumor entities [35,36]. In SCLC, the prognostic value of such scores is unclear, as various studies have shown controversial results [37,38]. A recent meta-analysis analyzing different inflammatory scores in patients with SCLC showed that only NLR was significantly associated with prognosis [19]. This result was confirmed in another meta-analysis [20]. We could not confirm this finding. This might be related to the cut-off chosen. We chose a cut-off of 4 in agreement with numerous published data. The use of data-dependent cut-off values in prognostic scores carries a risk of bias, especially when retrospectively associated with survival. We therefore decided against analyzing other NLR cut-off values that could have led to confirmation of their prognostic significance. In another retrospective, monocentric analysis, the authors propose a prognostic nomogram [39]. In this nomogram, NLR is included as a continuous variable and was found to be an independent prognostic factor for OS in a multiparametric analysis.

The mGPS is the only inflammation score that does not take lymphocyte count into account. Several studies have shown its prognostic significance in patients with SCLC [25,40,41,42,43], as we also demonstrated in our analysis. In a comparative retrospective study, different inflammation-based prognostic scores were compared [18]. Multivariate analysis showed that mGPS, ALI, prognostic nutritional index (PNI), CRP/albumin ratio, and albumin/globulin ratio were the most important determinants of prognosis. Due to the small number of cases, the mGPS could not be included in the meta-analysis discussed [19]. An important finding of this meta-analysis is that all relevant studies on this question were retrospective. Therefore, even considering the results of our retrospective work, a relevant conclusion is that there is a need for high-quality prospective studies on this question.

All prognostic scores discussed here do not consider tumor markers that are commonly used in clinical practice, such as neuron-specific enolase (NSE) or pro-gastrin-releasing peptide (ProGRP). We also did not include these tumor markers in our analysis.

A strength of this study is that it is based on a large, unselected, real-world population of patients with SCLC treated at an academic center in Switzerland. In addition, this study is based on high-resolution data with a very small number of missing values. The limitations of this study primarily relate to the retrospective data collection and the relatively small number of patients. We cannot exclude the possibility that the patient cohort studied here does not correspond to a patient cohort at a non-academic center, and it is possible that the assignment of patients to an academic center is subject to some selection. Validation of our results in an independent patient cohort would be necessary to confirm our findings. On the other hand, the patient characteristics in our study are consistent with those of other real-world cohorts [32,44,45]. Another limitation of our study is the time frame (2000–2010) of data collection. Recently, two randomized phase III trials have shown that the addition of a PD-L1 inhibitor to platinum-based combination chemotherapy in the first-line setting improves survival [9,10]. This has led to a new standard of care for patients with ED-SCLC that was not included in our analysis. The absolute benefit of combined chemo-immunotherapy is small, and considering the few patients who achieve long-term disease control despite this new therapeutic option, establishing validated and easily ascertainable prognostic and predictive factors is a major challenge for the future to make SCLC treatment more effective and individualized. To this end, prospective analysis and evaluation of potential prognostic factors in trials investigating new therapeutic approaches is of great importance.

## 5. Conclusions

In this real-world analysis of various clinical and laboratory parameters and different prognostic scores, we were able to validate only a few clinical (age, initial tumor stage) and a single laboratory (serum levels of AP) parameter as independent prognostic factors in a multivariate analysis. Of the numerous established prognostic scores, only the Manchester score and GPS showed a correlation with prognosis. We therefore recommend the use of simple clinical and laboratory factors instead of complex scores to estimate the prognosis of patients with SCLC.

## Figures and Tables

**Figure 1 cancers-14-02625-f001:**
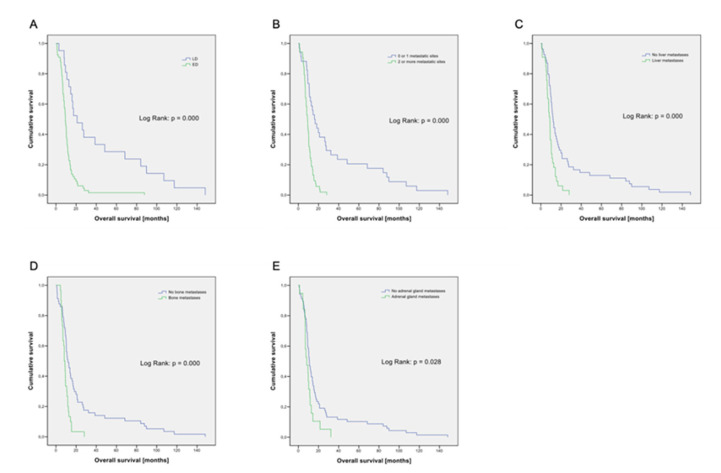
(**A**) Kaplan–Meier graph for overall survival according to staging. N = 87. LD = limited disease (n = 21); ED = extensive disease (n = 66). (**B**) Kaplan–Meier graph for overall survival according to number of metastatic sites. N = 87; 0 or 1 metastatic site: n = 34; 2 or more metastatic sites: n = 53. (**C**) Kaplan–Meier graph for overall survival according to liver metastases. N = 87. No liver metastases: n = 54; liver metastases: n = 33. (**D**) Kaplan–Meier graph for overall survival according to bone metastases. N = 87. No bone metastases: n = 57; bone metastases: n = 30. (**E**) Kaplan–Meier graph for overall survival according to adrenal gland metastases. N = 87. No adrenal gland metastases: n = 68; adrenal gland metastases: n = 19.

**Figure 2 cancers-14-02625-f002:**
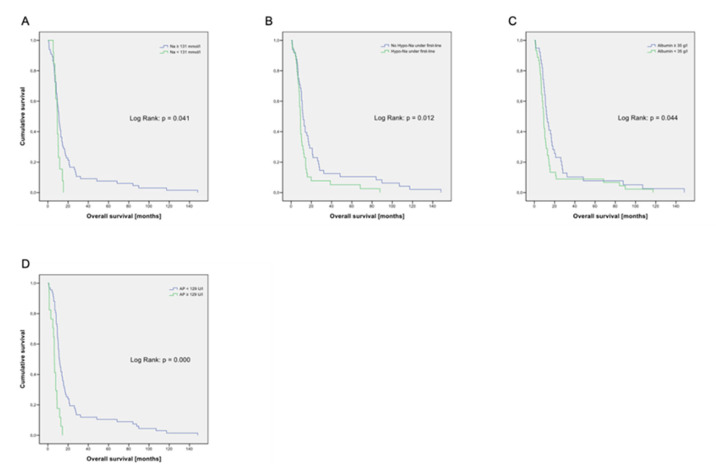
(**A**) Kaplan–Meier graph for overall survival according to sodium level. N = 79. Na = sodium (natrium). Na ≥ 131 mmol/L: n = 66; Na < 131 mmol/L: n = 13. (**B**) Kaplan–Meier graph for overall survival according to hyponatremia. N = 87. No Hypo-Na = no hyponatremia: n = 48; hypo-Na = hyponatremia: n = 39. (**C**) Kaplan–Meier graph for overall survival according to Albumin level. N = 84. Albumin ≥ 35 g/L: n = 39; Albumin < 35 g/L: n = 45. (**D**) Kaplan–Meier graph for overall survival according to alkaline phosphatase level. N = 84. AP < 129 U/L: n = 67; AP ≥ 129: n = 17.

**Figure 3 cancers-14-02625-f003:**
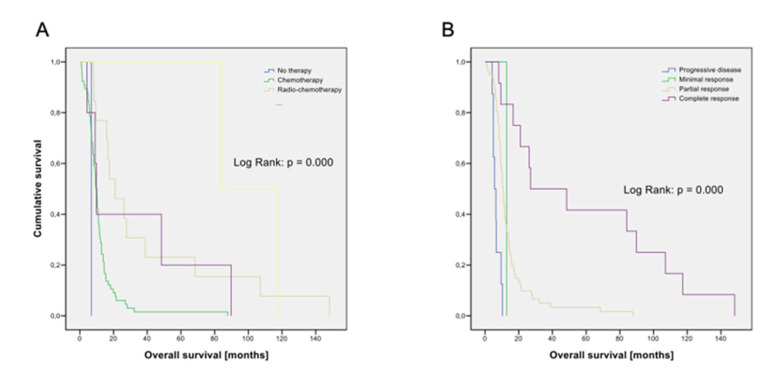
(**A**) Kaplan–Meier graph for overall survival according to first-line therapy. N = 87. No therapy: n = 1; chemotherapy: n = 66; radio-chemotherapy: n = 13; surgery and chemotherapy: n = 5; surgery and radio-chemotherapy: n = 2. (**B**) Kaplan–Meier graph for overall survival according to Response to first-line treatment. N = 82. Progressive disease: n = 8; minimal response: n = 1; partial response: n = 61; complete response: n = 12.

**Figure 4 cancers-14-02625-f004:**
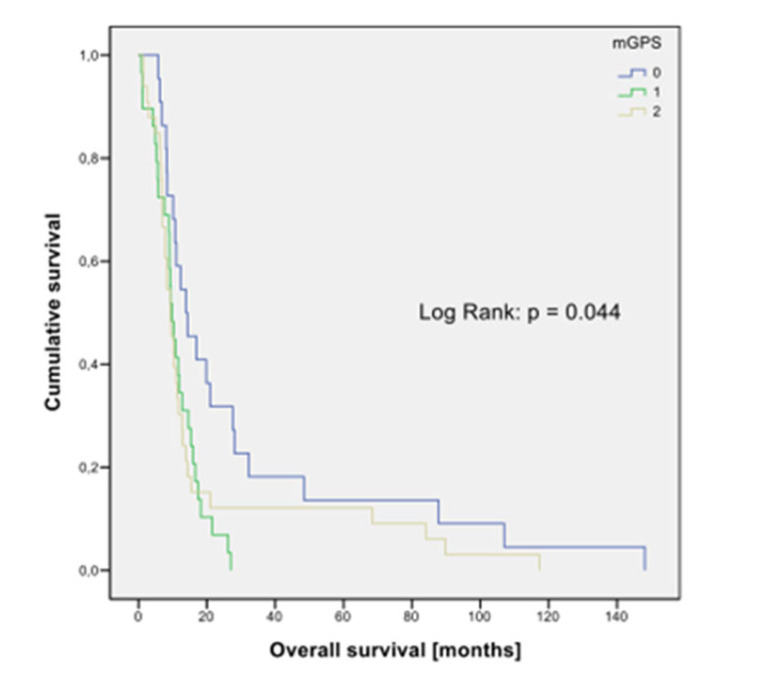
Kaplan–Meier graph for overall survival according to mGPS. N = 84; 0: n = 22; 1: n = 29; 2: n = 33.

**Table 1 cancers-14-02625-t001:** Univariate analysis of clinical parameters at baseline.

Variable	Categories	No. of Patients	MST (Month)	χ^2^	*p*-Value (Logrank)
Age (n = 87)					
	<65 years	50	9.429	0.083	0.774
	≥65 years	37	10.349		
Gender (n = 87)					
	Male	57	10.349	3.504	0.061
	Female	30	10.251		
Smoking (n = 82)					
	<50 pack years	41	9.429	0.151	0.698
	≥50 pack years	41	11.006		
Staging (n = 87)					
	LD	21	20.928	20.924	**<0.001**
	ED	66	9.232		
ECOG performance status (n = 48)					
	0 + 1	39	10.251	2.621	0.105
	2 + 3	9	8.936		
Weight (n = 63)					
	Normal weight (BMI ≥ 18.5 kg/m^2^)	56	10.382	0.134	0.714
	Underweight (BMI < 18.5 kg/m^2^)	7	9.528		
Weight loss during first-line therapy (n = 55)					
	<5%	47	11.006	1.362	0.243
	≥5%	8	10.382		
No. of metastatic sites (n = 87)					
	0 + 1	34	16.000	23.208	**<0.001**
	≥2	53	9.068		
Liver metastases (n = 87)					
	No	54	11.959	13.361	**<0.001**
	Yes	33	8.936		
Brain metastases (n = 87)					
	No	70	10.908	3.363	0.067
	Yes	17	10.119		
Bone metastases (n = 87)					
	No	57	11.959	12.439	**<0.001**
	Yes	30	8.312		
Adrenal gland metastases (n = 87)					
	No	68	10.908	4.798	**0.028**
	Yes	19	8.279		
Pleural effusion (n = 87)					
	No	67	10.251	0.824	0.364
	Yes	20	10.349		

Abbreviations: No. = number; MST = median survival time; LD = limited disease; ED = extensive disease; ECOG = European Cooperative Oncology Group; BMI = body mass index.

**Table 2 cancers-14-02625-t002:** Univariate analysis of laboratory parameters at baseline.

Variable	Categories	No. of Patients	MST (Month)	χ^2^	*p*-Value (Logrank)
Serum Na at baseline (n = 79)					
	≥131 mmol/L	66	10.908	4.180	0.041
	<131 mmol/L	13	9.232		
Hyponatremia under first-line therapy (n = 87)					
	≥131 mmol/L	48	11.630	6.342	0.012
	<131 mmol/L	39	9.232		
Serum Ca under first-line therapy (n = 78)					
	≤2.65 mmol/L	71	10.710	0.080	0.778
	>2.65 mmol/L	7	6.867		
Hypercalcemia at anytime during disease (n = 78)					
	≤2.65 mmol/L	67	10.710	0.143	0.706
	>2.65 mmol/L	11	7.786		
Serum Albumin (n = 84)					
	≥35 g/L	39	12.287	4.053	0.044
	<35 g/L	45	9.232		
Serum AP (n = 84)					
	<129 U/L	67	11.368	22.322	**<0.001**
	≥129 U/L	17	6.472		
Serum LDH (n = 76)					
	≤225 U/L	36	12.287	3.334	0.068
	>225 U/L	40	9.068		
Serum Bicarbonate (n = 53)					
	21–26 mmol/Lduring therapy 1	27	10.908	3.985	0.136
	<21 mmol/L during therapy 1	9	9.429		
	>26 mmol/Lduring therapy 1	17	9.528		
CRP (n = 87)					
	<10.0 mg/L	36	10.710	1.643	0.200
	≥10.0 mg/L	51	10.185		
Neutrophiles (n = 80)					
	≤6.700 × 10^9^/L	48	11.368	0.504	0.478
	>6.700 × 10^9^/L	32	10.119		
Leucocytes (n = 86)					
	≤10.00 × 10^9^/L	56	10.908	1.243	0.265
	>10.00 × 10^9^/L	30	9.528		

Abbreviations: No. = number; MST = median survival time; Na = natrium (sodium); Ca = calcium; AP = alkaline phosphatase; LDH = lactate dehydrogenase; CRP = c-reactive protein.

**Table 3 cancers-14-02625-t003:** Univariate analysis of therapy parameters at baseline.

Variable	Categories	No. of Patients	MST (Month)	χ^2^	*p*-Value (Logrank)
First-line treatment (n = 87)					
	No Therapy	1	6.867	22.241	**<0.001**
	Chemotherapy	66	9.528		
	Radio-chemotherapy	13	20.928		
	Surgery and chemotherapy	5	9.856		
	Surgery and radio-chemotherapy	2	84.074		
No. of days till first response (n = 68)					
	<50 days	54	11.368	0.172	0.678
	≥50 days	14	11.006		
Response to initial chemotherapy (n = 82)					
	Complete response	12	26.973	37.185	**<0.001**
	Partial response	61	10.710		
	Minimal response	1	12.780		
	Progressive disease	8	5.585		
Prophylactic cranial irradiation (n = 87)					
	Yes	15	9.429	0.910	0.340
	No	72	10.349		
Prophylactic cranial irradiation in patients with ED (n = 66)					
	Yes	7	26.973	3.221	0.073
	No	59	10.382		
Prophylactic cranial irradiation in patients with LD (n = 21)					
	Yes	8	17.413	1.356	0.244
	No	13	48.394		
Second-line chemotherapy (n = 87)					
	Yes	40	11.959	0.032	0.858
	No	47	8.936		
Second-line chemotherapy out of the patients who had a relapse (n = 64)					
	Yes	40	11.959	2.166	0.141
	No	24	9.035		

Abbreviations: No. = number; MST = median survival time; LD = limited disease; ED = extensive disease.

**Table 4 cancers-14-02625-t004:** Univariate analysis of staging systems.

Prognostic Index with Categories		No. of Patients	MST	χ^2^	*p*-Value (Logrank)
ALI (n = 50) BMI × Serum Albumin/Neutrophil-Lymphocyte ratio					
	≥19.5	30	11.039	0.657	0.418
	<19.5	20	9.856		
GPS (n = 84) One point for CRP > 10 mg/L or Albumin < 35 g/L					
	0	22	13.930	6.227	**0.044**
	1	29	9.856		
	2	33	9.232		
mGPS (n = 84) Like GPS, but 1 point only if CRP > 10 mg/L					
	0	34	10.710	1.540	0.463
	1	17	11.630		
	2	33	9.232		
London Group (n = 56)					
	Good prognosis - Karnofsky > 70 (=ECOG 0 + 1) - Na > 136 mmol/L - Alb > 39 g/L - AP less than 150% above the upper limit for laboratory)	7	12.287	2.470	0.291
	Intermediate prognosis	17	10.185		
	Poor prognosis: - Karnofsky < 40 (=ECOG 4 + 5) - Na < 135 mmol/L and Alb < 38 g/L - AP > 3 times upper limit for laboratory	32	9.232		
London Group with actualised boundary values (n = 48)					
	Good prognosis - Karnofsky > 70 (=ECOG 0 + 1) - Na ≥ 131 mmol/L - Alb ≥ 35 g/L - AP less than 150% above the upper limit for laboratory)	11	16.657	5.538	0.063
	Intermediate prognosis	26	9.265		
	Poor prognosis: - Karnofsky < 40 (=ECOG 4 + 5) - Na < 131 mmol/L and Alb < 35 g/L - AP > 3 times upper limit for laboratory	11	7.786		
Manchester scoring system (n = 33) 1 point per item: - LDH > 450 U/L - ED - Na < 132 mmol/L - Karnofsky < 60 (=ECOG 3 − 5) - AP > 165 U/L - Bicarbonate < 24 mmol/L					
	Good prognosis: 0 + 1 points	12	12.846	9.687	**0.008**
	Intermediate prognosis: 2 + 3 points	16	6.637		
	Poor prognosis: ≥4 points	5	5.749		
Manchester scoring system with actualised threshold values (n = 41)					
1 point per item: - LDH > 225 U/L - ED - Na < 131 mmol/L - Karnofsky < 60 (=ECOG 3 − 5) - AP ≥ 129 U/L - Bicarbonate < 21 mmol/L					
	Good prognosis: 0 + 1 points	9	16.657	8.828	**0.012**
	Intermediate prognosis: 2 + 3 points	27	8.936		
	Poor prognosis: ≥4 points	5	7.786		
Neutrophil-lymphocyte rate (n = 68)					
	Low (<4)	37	11.039	0.182	0.670
	High (≥4)	31	10.382		
New prognostic index (n = 42) 1 point per item: - LDH > 225 U/L - Albumin ≤ 34 g/L - Neutrophils > 7.5 × 10^9^/L - ED - ECOG > 1					
	Good prognosis: 0 + 1 points	8	12.287	4.846	0.089
	Intermediate prognosis: 2 + 3 points	23	10.185		
	Poor prognosis: 4 + 5 points	11	6.472		
PNI (n = 70) 10 × Albumin g/dL + 0.005 × Lymphocytes/mm^3^					
	<52.48	60	10.710	3.616	0.057
	≥52.48	6	15.310		

Abbreviations: No. = number; MST = median survival time; ALI = advanced lung cancer inflammation index; BMI = body mass index, GPS = Glasgow Prognostic Score; CRP = C-reactive protein; mGPS = modified Glasgow Prognostic Score; ECOG = European Cooperative Oncology Group; Na = natrium; Alb = albumin; AP = alkaline phosphatase; LDH = lactate dehydrogenase; ED = extensive disease; PNI = prognostic nutritional index.

**Table 5 cancers-14-02625-t005:** Baseline characteristics and their correlation to long-term survival.

Variable and Categories	Survival < 2 Years	Survival ≥ 2 Years	*p*-Value (χ^2^ Test)
Age (n = 88)			
<65 years	42	9	0.860
≥65 years	31	6	
Gender (n = 88)			
Male	51	7	0.084
Female	22	8	
Smoking (n = 83)			
<50 pack years	35	7	0.779
≥50 pack years	35	6	
Staging (n = 88)			
LD	11	11	**<0.001**
ED	62	4	
ECOG performance status (n = 49)			
0 + 1	32	8	0.142
2 + 3	9	0	
Weight (n = 64)			
Normal weight (BMI ≥ 18.5 kg/m^2^)	46	11	0.748
Underweight (BMI < 18.5 kg/m^2^)	6	1	
Weight loss during first-line therapy (n = 56)			
<5%	36	12	0.111
≥5%	8	0	
No. of metastatic sites (n = 88)			
0+1	21	14	**<0.001**
≥2	52	1	
Liver metastases (n = 88)			
No	41	14	**0.007**
Yes	32	1	
Brain metastases (n = 88)			
No	56	15	**0.037**
Yes	17	0	
Bone metastases (n = 88)			
No	44	14	**0.014**
Yes	29	1	
Adrenal gland metastases(n = 88)			
No	55	14	0.123
Yes	18	1	
Pleural effusion (n = 88)			
No	54	14	0.103
Yes	19	1	

Abbreviations: LD = limited disease; ED = extensive disease; ECOG = European Cooperative Oncology Group; BMI = body mass index; No. = number.

**Table 6 cancers-14-02625-t006:** Baseline laboratory parameters and their correlation to long-term survival.

Variable and Categories	Survival < 2 Years	Survival ≥ 2 Years	*p*-Value (χ^2^ Test)
Serum Na at baseline (n = 80)			
≥131 mmol/L	55	12	0.098
<131 mmol/L	13	0	
Hyponatremia under first-line therapy (n = 88)			
≥131 mmol/L	37	12	**0.037**
<131 mmol/L	36	3	
Serum Ca under first-line therapy (n = 79)			
≤2.65 mmol/L	62	9	0.064
>2.65 mmol/L	5	3	
Hypercalcemia at anytime during disease (n = 79)			
≤2.65 mmol/L	58	9	0.304
>2.65 mmol/L	9	3	
Serum Albumin (n = 85)			
≥35 g/L	30	10	**0.046**
<35 g/L	41	4	
Serum AP (n = 85)			
<129 U/L	54	14	**0.041**
≥129 U/L	17	0	
Serum LDH (n = 76)			
≤225 U/l	28	8	0.145
>225 U/l	36	4	
Serum Bicarbonate (n = 53)			
21–26 mmol/L during therapy 1	22	5	0.474
<21 mmol/L during therapy 1	8	1	
>26 mmol/L	16	1	
CRP (n = 88)			
<10.0 mg/L	28	9	0.122
≥10.0 mg/L	45	6	
Neutrophil (n = 81)			
≤6.700 × 10^9^/L	39	10	0.588
>6.700 × 10^9^/L	27	5	
Leucocytes (n = 87)			
≤10.00 × 10^9^/L	46	11	0.484
>10.00 × 10^9^/L	26	4	

Abbreviations: Na = natrium; Ca = calcium; AP = alkaline phosphatase; LDH = lactate dehydrogenase; CRP = C-reactive protein.

**Table 7 cancers-14-02625-t007:** Therapy parameters and their correlation to long-term survival.

Variable and Categories	Survival < 2 Years	Survival > 2 Years	*p*-Value (χ^2^ Test)
First-line treatment (n = 88)			
No Therapy	1	0	**<0.001**
Chemotherapy	62	4	
Radio-chemotherapy	7	7	
Surgery and chemotherapy	3	2	
Surgery and radio-chemotherapy	0	2	
No. of days till first response (n = 69)			
<50 days	43	12	0.210
≥50 days	13	1	
Response to initial chemotherapy (n = 83)			
Complete response	4	9	**<0.001**
Partial response	55	6	
Minimal response	1	0	
Progressive disease	8	0	
Prophylactic cranial irradiation (n = 88)			
Yes	62	10	0.095
No	11	5	
Prophylactic cranial irradiation in patients with ED (n = 66)			
Yes	56	3	0.335
No	6	1	
Prophylactic cranial irradiation in patients with LD (n = 22)			
Yes	6	7	0.665
No	5	4	
Second-line chemotherapy (n = 88)			
Yes	35	5	0.301
No	38	10	
Second-line chemotherapy out of the patients who had a relapse (n = 64)			
Yes	35	5	0.605
No	22	2	

Abbreviations: No. = number; ED = extensive disease; LD = limited disease.

**Table 8 cancers-14-02625-t008:** Multivariate analysis.

Variable and Categories	Relative Risk	95% CI	*p*-Value
Age			
<65 vs. ≥65 years	2.103	1.135–3.898	**0.018**
Gender			
Male vs. female	0.785	0.408–1.512	0.470
Smoking, py			
<50 vs. ≥50 pack years	0.923	0.515–1.655	0.789
Staging			
LD vs. ED	2.684	1.153–6.251	**0.022**
No. of metastases			
(0 + 1) vs. (≥2)	1.332	0.628–2.824	0.455
Hyponatremia under first-line therapy			
No vs. Yes	0.913	0.411–2.028	0.823
Albumin < 35 g/L			
No vs. Yes	1.598	0.898–2.841	0.111
AP > 129 U/L			
No vs. Yes	3.311	1.471–7.452	**0.004**
LDH > 225 U/L			
No vs. Yes	0.829	0.474–1.452	0.512

Abbreviations: py = pack years; LD = limited disease; ED = extensive disease; No. = number; AP = alkaline phosphatase; LDH = lactate dehydrogenase.

## Data Availability

The data presented in this study are available on request from the corresponding author.

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
