# Peer review of "Validation of Pretreatment Prognostic Factors and Prognostic Staging Systems for Small Cell Lung Cancer in a Real-World Data Set"

_cancers, 2022, doi:10.3390/cancers14112625_

Round 1

Reviewer 1 Report

Please consider the following comments and seriously well reorganize the manuscript before acceptance for publication.

  1. Line 153, there is no space before and after “=”, while in Table 1A, there is a space before and after “=”. Please homogenous the format throughout the manuscript.
  2. Please use italic p as it refers to a p-value, for example, line 191, lines 209-212. Please check throughout the whole manuscript.
  3. Please check the figures and tables seriously. For example, Table 4 is spitted into two pages. The title of each figure is too far away from its figure.
  4. The X-axis for all the figures: “120,00” is twelve thousand, “120.00” is one hundred and twenty; Please use “.”. For the Y-axis, I’d suggest the authors use 100% as the scale and the same as the X-axis, please use “.”.
  5. Please add at least a simple legend for each figure including abbreviation, and patient numbers for easier tracking. I’d suggest the authors reorganize the figures, for example, put Figure 1A-1E together with only one figure title to make them well organized.

Author Response

Please see letter to the editor. Including a point-by-point reply to all reviewers' comments.

Reviewer 2 Report

Thank you for your work, yet there are some flaws to be addressed:

  • the paper is full of misprints, please correct the text.
  • the manuscript should be revised by a native English speaker
  • The authors should highlight better the impact of their research on clinical practice
  • the main limit is the retrospective nature of the study
  • A consistent background is missing in the introduction section. I suggest the authors to read the following papers PMID: 32849916  PMID: 34646363   

Author Response

Please see letter to the editor. Including a point-by-point reply to all reviewers' comments.

This manuscript is a resubmission of an earlier submission. The following is a list of the peer review reports and author responses from that submission.

Round 1

Reviewer 1 Report

Thank you for your valuable work, however there are some flaws to be addressed.

  • The figures are too small to be presented. Please improve the quality of the images.
  •  All the tables are not correctly labelled, please correct it
  • All the tables must include a legend of abbreviations used
  • One of the main limit of this research, it is not have considered the combo chemo-IO that is currently the standard-of-care in this disease.suggest you to implement the discussion and the limitation section with a special focus on that. Please read this paper which can be useful doi.org/10.1177/17588359211018018

Reviewer 2 Report

Although SCLC qualifies as a recalcitrant disease, the prognosis of SCLC is not uniformly poor, and some clinical variables are paramount prognostic determinants and key eligibility criteria in predicting the overall survival of patients.

The study realized by Hagmann R and colleagues aimed at generating and validating various prognostic factors scores in an unselected real-world cohort of 92 patients diagnosed with LD-SCLC and ED-SCLC. They considered simple clinical and laboratory factors to estimate patients’ prognoses.

In fact, only age, initial tumor stage, and serum levels of alkaline phosphatase were validated as independent prognostic factors in a multivariate analysis. Of the numerous established prognostic scores, only the Manchester score and the Glasgow Prognostic Score showed a correlation with prognosis. On the other side, they didn’t confirm the prognostic role of most of the complex prognostic scores published.

Major comments:

  • Firstly, all of the data were obtained retrospectively, which made it susceptible to the inherent weaknesses of retrospective data collection, considering the monocentric nature of the development cohort.
  • The sample size of our validation cohort was not very large. Another external validation with a larger sample size of the predictive model is still necessary.
  • The lack of data on tumor markers associated with SCLC such as NSE, proGRP and other inflammation-related hematological markers both of which are key determinants of tumor survival surely doesn’t help clinicians in determining pretherapeutic risk factors.
  • I appreciate that the strength of this study is based on an unselected real-world population of patients diagnosed with SCLC treated. You recommended that using simple clinical and laboratory factors instead of complex scores could estimate the prognosis of patients with SCLC. I think that the evaluation of the simplified prognostic score in the real-life setting and in controlled trials, particularly at the time of the immunochemotherapy strategy emergence, could be very useful. But the nomogram that you built should be included such previously important information to efficiently increase your model accuracy.
  • In this way, I could have expected a validation of these scores in patients receiving immunochemotherapy for SCLC also based on specific genotypic characterization.
  • You must improve the linearity of tables but more importantly the resolution of all figures.

Minor comments:

  • Scientific content seems good but the English style and language used in the manuscript need to be revised. The methodology is sufficiently documented to allow replication studies and the choice of statistical test for Univariate and Multivariate analysis seems correct.

Reviewer 3 Report

The manuscript titled “Validation of pretreatment prognostic factors and prognostic staging systems for small cell lung cancer in a real-world data set” describes a simple method that the authors recommended using clinical and laboratory factors to prognosis the patient with small cell lung cancer other than the complex established prognostic scores. It’s a good idea to use a simple approach to solve the problem instead of the complicated one. However, each model with its limitations. The followings are some concerns and comments that have been pointed out that the authors may want to consider.

Major Concerns and Comments:

  1. Provide higher resolution Figure 1, Figure 2, Figure 3, and Figure 4. The details are too difficult to be distinguished.
  2. Lines 29-31: The authors recommended using simple clinical and laboratory factors to estimate the prognosis of SCLC patients instead of the published complex prognostic scores that most of them could not be confirmed. It’s good to use a simple method instead of a complex one if it works perfectly. In this manuscript, the authors selected 92 SCLC patients’ data from Jan/2000 to Dec/2010. Would you please explain why the data did not cover 2011 to 2021? Additionally, I’d suggest the authors use some SCLC patients’ data to verify the prognosis of SCLC that is recommended in this paper to confirm it better than the published complex prognostic scores.
  3. Line 40: “Most patients with SCLC have a history of tobacco smoking.” While in Table 1A smoking parameter p=0.698; Table 3A smoking parameter p=0.779; Table 4 smoking parameter p=0.789. Provide the reason why smoking seems to have no effect?
  4. I’d highly suggest the authors at least describe those close to significant differences as well, instead of only describing those with significant differences (p<0.05). For example, Table 1A, gender parameter p=0.061 and brain metastases parameter p=0.067; Table 1B, serum LDH parameter p=0.068; Table 1C, prophylactic cranial irradiation in patients with ED parameter p=0.073. The same to Table 3A, 3B, 3C.
  5. Table 2: There are 10 p values in this table, 4 of them range from 0.291 to 0.670; 3 of them less than 0.05; while the left 3 with p-value 0.063, 0.089, and 0.057, respectively. Please add some descriptions and/or discussions at least to those 3 close to 0.05 instead of only focusing on the 0.05 cutoff value.
  6. Table 1A gender parameter p=0.061 very close to 0.05 significant difference cutoff value, and Table 3A gender parameter p=0.084. Please explain and discuss this analysis result.
  7. Different model has their advantage and of course their limitations. Please provide a detailed comparison of advantages and disadvantages between established complex prognostic scores and the authors’ recommended simple clinical/laboratory factors. What it can do and what it can’t prognosis. I’d suggest the authors add a summary table to make it clearer.

Minor Concerns and Comments:

  1. Line 11: There is an extra “s” between “and” and “single”.
  2. I’d suggest the authors add a space before the reference “[]”. Please check throughout the whole manuscript.
  3. Line 92: There is an extra space before the word “age”. There are a lot of extra spaces, please check throughout the whole manuscript.
  4. Please use italic p as it refers to a p-value. Please check throughout the whole manuscript.
  5. Line 140: I’d suggest the authors add a serial number to the 13 parameters in Table 1A to make it clearer. The same to Table 1B, Table 1C.
  6. Table 4: “Nr. of metastases”, please define “Nr.” before using it.
